# Slug Mediates MRP2 Expression in Non-Small Cell Lung Cancer Cells

**DOI:** 10.3390/biom12060806

**Published:** 2022-06-09

**Authors:** Xieyi Zhang, Wangyang Liu, Kazue Edaki, Yuta Nakazawa, Saori Takahashi, Hiroki Sunakawa, Kenta Mizoi, Takuo Ogihara

**Affiliations:** 1Laboratory of Biopharmaceutics, Department of Pharmacology, Faculty of Pharmacy, Takasaki University of Health and Welfare, 60 Nakaorui-chou, Takasaki-shi 370-0033, Gunma, Japan; k-edaki@takasaki-u.ac.jp (K.E.); 1721056@takasaki-u.ac.jp (Y.N.); s-takahashi@takasaki-u.ac.jp (S.T.); mizoi-k@takasaki-u.ac.jp (K.M.); togihara@takasaki-u.ac.jp (T.O.); 2Laboratory of Clinical Pharmacokinetics, Graduate School of Pharmaceutical Sciences, Takasaki University of Health and Welfare, 60 Nakaorui-machi, Takasaki-shi 370-0033, Gunma, Japan; 2120104@takasaki-u.ac.jp (W.L.); 1521052@takasaki-u.ac.jp (H.S.)

**Keywords:** efflux transporter, transcriptional regulator, scaffold protein, non-small-cell lung cancer, SNAI family

## Abstract

Transcriptional factors, such as Snail, Slug, and Smuc, that cause epithelial-mesenchymal transition are thought to regulate the expression of Ezrin, Radixin, and Moesin (ERM proteins), which serve as anchors for efflux transporters on the plasma membrane surface. Our previous results using lung cancer clinical samples indicated a correlation between Slug and efflux transporter MRP2. In the current study, we aimed to evaluate the relationships between MRP2, ERM proteins, and Slug in lung cancer cells. HCC827 cells were transfected by Mock and Slug plasmid. Both mRNA expression levels and protein expression levels were measured. Then, the activity of MRP2 was evaluated using CDCF and SN-38 (MRP2 substrates). HCC827 cells transfected with the Slug plasmid showed significantly higher mRNA expression levels of MRP2 than the Mock-transfected cells. However, the mRNA expression levels of ERM proteins did not show a significant difference between Slug-transfected cells and Mock-transfected cells. Protein expression of MRP2 was increased in Slug-transfected cells. The uptake of both CDCF and SN-38 was significantly decreased after transfection with Slug. This change was abrogated by treatment with MK571, an MRP2 inhibitor. The viability of Slug-transfected cells, compared to Mock cells, significantly increased after incubation with SN-38. Thus, Slug may increase the mRNA and protein expression of MRP2 without regulation by ERM proteins in HCC827 cells, thereby enhancing MRP2 activity. Inhibition of Slug may reduce the efficacy of multidrug resistance in lung cancer.

## 1. Introduction

Non-small cell lung cancer (NSCLC), which accounts for the majority of lung cancer cases, has a 5-year survival rate as low as 20% [1]. This rate is notably reduced in patients with metastasis as surgery becomes increasingly difficult [2]. Metastasis to the brain, for example, occurs more frequently in lung cancer patients, and many surgeries are unavailable [3]. Thus, although chemotherapy is often used as a first-line treatment [4], it can be influenced by multidrug resistance, resulting in a poor prognosis [5].

P-glycoprotein (P-gp, ATP-binding cassette subfamily B member, ABCB1), multidrug resistance-associated protein2 (MRP2, ABCC2), and breast cancer resistance protein (BCRP, ABCG2) are drug efflux transporters that cause multidrug resistance [6]. They lead to the reduced accumulation of anticancer drugs in cells [7]. P-gp functions as an efflux pump for a wide range of amphiphilic, bulky type II cationic drugs and other hydrophobic compounds such as endogenous and exogenous metabolites or toxins, steroid hormones, hydrophobic peptides, and glycolipids [8]. MRP2 is suggested to be closely associated with cisplatin resistance in NSCLC [9,10].

Epithelial-mesenchymal transition (EMT) is the first step in the conversion of primary epithelial cells into mesenchymal cells, which are involved in cancer metastasis and are considered to be associated with increased malignancy [11]. EMT is also considered a factor in multidrug resistance during anticancer therapies [12]. SNAI family members, including Snail, Slug, and Smuc, are transcriptional regulators that trigger EMT [13]. Recently, scaffold proteins, including ezrin (Ezr), radixin (Rdx), and moesin (Msn) (ERM proteins), have been shown to regulate the location of efflux transporters on the cell membrane, consequently regulating their functional activity [7]. Our previous research showed that overexpression of Snail induced an increase in P-gp activity in HCC827 lung cancer cells, which was suppressed by knockdown of Msn. Thus, Snail-induced EMT may regulate Msn expression and P-gp efflux activity in the HCC827 cell line [14]. 

In our previous study, surgically excised lung tissues, including cancer tissues and adjacent non-cancer tissues, were obtained from nine patients. We extracted mRNA from these tissues and the mRNA expression levels were measured. A high correlation between relative (cancer/noncancer tissue) mRNA expression levels of Snail and Msn and Msn and P-gp was found in clinical lung cancer and noncancer samples [15]. These results are consistent with those of the in vitro study [14].

Moreover, the mRNA expression levels of Slug have also been found to correlate with MRP2 [15]. In the current study, we aimed to evaluate whether the relationships between Slug and MRP2 and related scaffold proteins can be detected in lung cancer cells.

## 2. Materials and Methods

### 2.1. Materials

Mock and Slug-encoding plasmids were constructed using Vectorbuilder (Chicago, IL, USA) and contained human Stuffer_300bp (Mock) or SNAI2 (NM_003068.5, Slug), respectively, and enhanced green fluorescent protein sequences. 5(6)-Carboxy-2′,7′-dichlorofluorescein (CDCF, substrate of MRP2) was purchased from Promokine (Heidelberg, Germany). 7-Ethyl-10-hydroxycamptothecin (SN-38) was purchased from Sigma-Aldrich (St. Louis, MO, USA). Mouse anti-human GAPDH (sc-365062), Slug (sc-166476), Msn (sc-58806), goat anti-mouse IgG-HRP (sc-2005), and goat anti-rabbit IgG-HRP (sc-2004) antibodies were purchased from Santa Cruz Biotechnology (Dallas, TX, USA). MRP2 (ab-3373), Ezr (T567), and Rdx (PAS-21660) were purchased from Abcam (Cambridge, UK), Cell Signaling Technology (Danvers, MA, USA), and Invitrogen (Waltham, MA, USA), respectively. All other reagents were commercial reagent-grade products.

### 2.2. Cell Culture

The NSCLC cell line HCC827 was obtained from the American Type Culture Collection (Manassas, VA, USA). Cells were cultured in Dulbecco’s modified Eagle’s medium (Wako, Tokyo, Japan) (low glucose) supplemented with 10% fetal bovine serum (Biosera, Manila, Philippines), 100 units/mL penicillin (Sigma-Aldrich, St. Louis, MO, USA), and 0.1% mg/mL streptomycin. Cells were seeded onto 24-well plates at a density of 1.4 × 10^5^ cells/well at 37 °C in a humidified atmosphere of 5% CO_2_ in air.

### 2.3. Transfection of Slug and Mock Plasmids

According to the product manual, 1.4 × 10^5^ Mock or Slug plasmids were transfected into HCC827 cells using Lipofectamine 3000 Transfection Reagent (Invitrogen, Waltham, MA, USA) the day after seeding in 24-well plates. Cells were 80–90% confluent before transfection. DNA plasmid (500 ng) together with 1.5 μL Lipofectamine 3000 Reagent were diluted in Opti-MEM medium and added to each well. The cells were incubated with the transfection reagent mixture for 6 h, followed by a change of the medium. The cells were used for further experiments two days after transfection.

### 2.4. mRNA Extraction and cDNA Synthesis

The mRNA was extracted from NSCLC cell lines seeded in 24-well culture plates using a previously reported method [14]. Total RNA was extracted two days post-transfection using NucleoSpin RNA Midi (Machery-Nagel, Düren, Germany), according to the manufacturer’s instructions. The cDNA was synthesized from 1 mg of total RNA using ReverTraAce (Toyobo, Osaka, Japan), according to the manufacturer’s instructions, using a T100™ Thermal Cycler. Primers used for quantification were as follows: GAPDH (forward, 5′-CCCTTAAGAGGGATGCTGCC-3′; reverse, 5′-TACGGCCAAATCCGTTCACA-3′), Ezr (forward, 5′-CACGCTTGTGTCTTTAGTGCTCC-3′; reverse, 5′-ACTCAGACTTTACAGGCATTTTCC-3′), Rdx (forward, 5′-TGCACCTCGTCTGAGAATCA-3′; reverse, 5′-CTCTAATTGTGCCCTTTCCAAC-3′), Msn (forward, 5′-GCCCTGGGTCTCAACATCTA-3′; reverse, 5′-GACGGCGCATGTATAGTTCA-3′), P-gp (forward, 5′-CCCATCATTGCAATAGCAGG-3′; reverse, 5′-GTTCAAACTTCTGCTCGTGA-3′), MRP2 (forward, 5′-AGCAGGTATTCGTTGGTTTTCT-3′; reverse, 5′-AACCAGGAGCCATGTGCCTA-3′), BCRP (forward, 5′-ACTGGCTTAGACTCAAGCACA-3′; reverse, 5′-ATAGGCCTCACAGTGATAACCA-3′), Slug (forward, 5′-TGCGATGCCCAGTCTAGAAA-3′;reverse, 5′-GAAAAGGCTTCTCCCCCGT-3′).

### 2.5. Quantitative Real-Time Polymerase Chain Reaction (qRT-PCR)

Quantitation of mRNA expression levels by qRT-PCR was performed with the Power SYBER™ Green PCR Master Mix (Applied Biosystems, Waltham, MA, USA) using the MX3000P™ Multiplex Quantitative PCR System (Stratagene, San Diego, CA, USA). The mRNA expression levels of the target genes were quantified relative to that of the housekeeping gene GAPDH using the 2^−ΔΔCT^ method.

### 2.6. Electrophoresis and Western Blotting

Whole-cell extracts were obtained from HCC827 cells transfected with Mock- or Slug-encoded plasmids. Two days after transfection, the cells seeded on 24-well culture plates were rinsed three times with ice-cold PBS and then lysed with RIPA buffer containing complete protease inhibitor cocktail (Roche, Basel, Switzerland) for 30 min on ice. The lysate was then centrifuged at 15,000× *g* for 30 min at 4 °C. The supernatant was mixed with sample buffer (250 mM Tris-HCl, pH 6.8, 10% SDS, 30% glycerol, 5% β-mercaptoethanol, and 0.02% bromophenol blue) and separated by electrophoresis on 4%–12% polyacrylamide gels (Bio-Rad Laboratories Inc., Hercules, CA, USA). The proteins were transferred onto a polyvinylidene difluoride membrane (GE Healthcare Life Sciences, Little Chalfont, UK). The membrane was blocked with 5% skimmed milk and incubated overnight at 4 °C with primary antibodies. The following day, the membrane was washed and incubated with goat anti-mouse IgG-HRP secondary antibody for 1 h at 25 °C. The bands were detected by LAS3000 (Fujifilm, Tokyo, Japan) using an ECL substrate (GE Healthcare Life Sciences), and GAPDH was used as a loading control. The parameter arbitrary to background (AU-BG) obtained from Multi Gauge V3.0 imaging analysis system (Fujifilm, Tokyo, Japan) was used for statistical analysis. 

### 2.7. CDCF Uptake Assay

Two days after transfection, cells were washed twice with ice-cold PBS and incubated with 5 μM CDCF for 1 h at 37 °C. For the inhibition of MRP2, MK571 was added 1 h before incubation with CDCF. After washing three times with cold PBS, NaOH (0.1 N) was applied. The cells were then transferred to a 96-well assay plate (Corning Inc., New York, NY, USA). The fluorescence was measured using a multilabel plate reader (PerkinElmer, Yokohama, Japan) with excitation and emission wavelengths of 485 and 538 nm. The cell-to-medium (C/M) ratio of CDCF was calculated using the following formula: (1)$$C/M\;\mathrm{ratio}\;\left(\mu L/\mathrm{mg}\right)=\frac{\mathrm{intracellular}\;\mathrm{concentration}\;\left(\mu \mathrm{mol}/L\right)}{\mathrm{extracellular}\;\mathrm{concentration}\;\left(\mu \mathrm{mol}/L\right)\times \;\mathrm{protein}\;\mathrm{concentration}\;\left(\mathrm{mg}/\mu L\right)}$$

### 2.8. SN-38 Uptake Assay

We chose SN-38, which is an active metabolite of irinotecan and an MRP2 substrate, to evaluate the transport activities of MRP2, according to our previous experiment [16]. Uptake activity was measured using cells 2 days after transfection. SN-38 uptake was measured by LC-MS/MS. Cells lysed with NaOH were used to measure protein concentration according to the Lowry method. As with the calculation of CDCF uptake, changes in the intracellular accumulation of drugs were determined using the C/M ratio.

### 2.9. Cell Viability Assay

The MTT assay was performed to assess cell viability in 24-well plates at 4 × 10^4^ cells/mL. Two days after transfection, 500 μL of 50 μM SN-38 or 1% DMSO in medium (control) was applied. After incubation at 37 °C for 1 h, the solution in the wells was aspirated, and 500 μL MTT was added. The cells were incubated with MTT at 37 °C for 30 min. Then, MTT was removed and DMSO was added at 1 mL/well. The plate was shaken for 5 min, then 200 μL of the cell suspensions were added to the 96-well plate, and the absorbance was measured at 560 nm.

### 2.10. Statistical Analysis

All results are presented as mean ± standard deviation (SD) (n = 3 or 4). Student’s *t*-test or Mann–Whitney test was used for comparisons between two groups. Holm’s test was used to compare more than two groups. Statistical significance was set at *p* < 0.05. 

## 3. Results

### 3.1. mRNA Expression Levels of Slug in Cells

Using fluorescence imaging, we confirmed that Mock- and Slug-encoding plasmids were successfully transfected into the HCC827 cell line. Slug mRNA expression and protein expression were significantly higher in Slug-transfected cells than in Mock-transfected cells (*p* < 0.05, Figure 1).

### 3.2. mRNA Expression Levels of Efflux Transporters and ERM Proteins 

To examine the effect of Slug on efflux transporters and ERM proteins, we quantified the mRNA levels in Slug- and Mock-transfected HCC827 cells. The mRNA expression level of MRP2 was significantly higher in Slug-overexpressing HCC827 cells than in Mock cells (*p* < 0.05); however, the expression levels of P-gp, BCRP, Ezr, Rdx, and Msn remained unchanged (not shown) (Figure 2).

### 3.3. Protein Expression Level of MRP2 in Slug-Overexpressing HCC827 Cells 

Western blotting was performed to confirm the protein expression level of MRP2. The protein expression level of MRP2 also significantly increased in Slug-transfected cells (*p* < 0.05, Figure 3). The protein expression levels of Ezr, Rdx, and Msn remained unchanged (not shown). Protein expression levels were consistent with mRNA expression levels.

### 3.4. Uptake of CDCF and SN-38 in Slug-Transfected Cells

HCC827 cells were incubated with the MRP2 substrate CDCF (5 μM). The C/M ratio of CDCF in Slug-transfected cells was significantly lower than that in Mock-transfected cells (*p* < 0.05). Moreover, this change was reversed by the MK571 treatment (Figure 4).

When cells were exposed to 2 μM SN-38, the C/M ratio of SN-38 was significantly decreased in Slug-transfected cells compared to that in Mock-transfected cells (*p* < 0.05). This change was reversed after pre-incubation with MK571 for 1 h (Figure 5).

### 3.5. Cell Viability in Slug-Transfected Cells after SN-38 Administration

We performed a cell viability assay to determine the toxicity of SN-38 in Slug-overexpressing HCC827 cells. In the Slug-transfected cells, cell viability increased significantly (*p* < 0.05, by 38.0%) compared to the Mock-transfected cells (by 25.7%) after incubation with 50 μM SN-38 for 1 h (Figure 6).

## 4. Discussion

In this study, we first confirmed that Slug-expressing HCC827 cells showed increased mRNA and protein expression of MRP2. We also performed an uptake assay of CDCF and SN-38 and a cell viability assay using SN-38 to evaluate the activity of MRP2. Moreover, increased activity of MRP2 can be inhibited by MK571, a potential inhibitor of MRP2. All these experiments revealed the increased activity of MRP2 in Slug-expressing HCC827 cells.

According to the current results, Slug may directly increase MRP2 activity in HCC827 lung cancer cells, without enhancing the activity of ERM proteins. Moreover, our previous clinical study suggested that Slug may contribute to the functional regulation of MRP2 without the regulation of ERM proteins in lung cancer cells [15]. In accordance with these results, we found that in vitro data are consistent with the data from clinical samples, which indicates that cell experiments may help predict clinical outcomes in lung cancer patients.

A high correlation between the relative mRNA expression levels of Snail and Msn and Msn and P-gp was also found in clinical lung cancer samples [15]. In our previous study using HCC827 cells, overexpression of Snail induced an increase in P-gp activity through an increase in Msn protein [14]. Moreover, MRP2 is often regulated by Rdx in liver cancer [17,18], breast cancer cell lines [19], and the gastrointestinal tract [20]. Furthermore, previous studies have shown that Rdx modulates MRP2 activity in A549 lung cancer cell lines [21,22]. Based on these results, we expected to observe the Slug/Rdx/MRP2 linkage system in lung cancer, similar to the Snail/Msn/P-gp linkage system. However, Slug directly increased MRP2 expression and activity, without Rdx intermediation. In other words, the mechanism by which Slug enhances MRP2 activity is different from that by which Snail enhances P-gp activity, although MRP2 and P-gp are both efflux transporters and Slug and Snail are both transcriptional regulators involved in EMT.

Our current study has several limitations. Firstly, we only considered SN-38 as a substrate of MRP2 during cell viability and SN-38 uptake assays. However, SN-38 is also a substrate of BCRP [23]. Moreover, MRP family members such as MRP1, MRP3, and MRP5 may also be involved in resistance to SN-38 administration [24]. It may also be a substrate of other unknown transporters which can also be regulated by Slug or EMT. Secondly, we only employed the HCC827 cell line for our current experiment. The results might be different when using other lung cancer cell lines or the cell line from other organs.

## 5. Conclusions

We concluded that Slug increased the mRNA and protein expression of MRP2 without regulation by ERM proteins in HCC827 cells, thereby enhancing the activity of MRP2. The inhibition of Slug may not only suppress EMT but also reduce the efficacy of MRP2 in lung cancer.

## Figures and Tables

**Figure 1 biomolecules-12-00806-f001:**
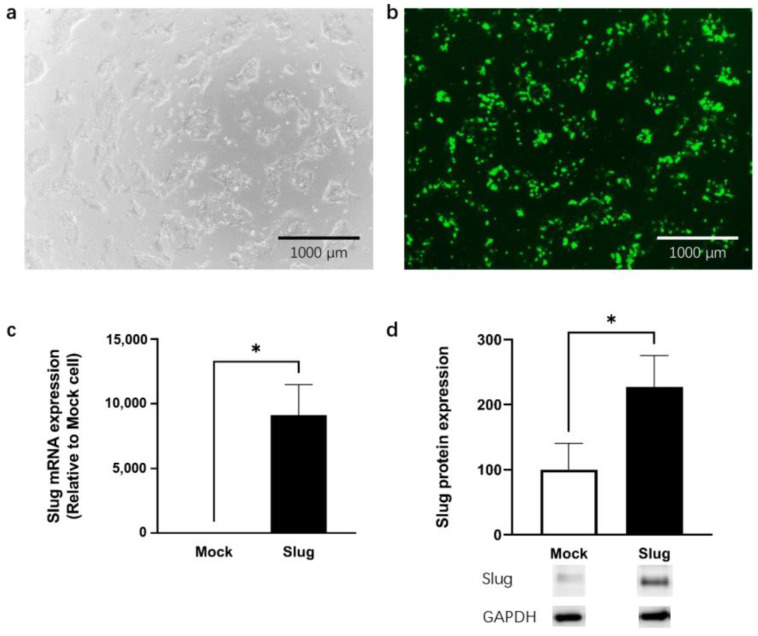
HCC827 cell line was successfully transfected with the Slug plasmid shown on white image (**a**) and fluorescence image (**b**). The results showed a significantly higher expression of Slug mRNA (**c**) and proteins (**d**) than that of the Mock cells (n = 4, * *p* < 0.05). qRT-PCR and Western blotting data represented the mean ± SD.

**Figure 2 biomolecules-12-00806-f002:**
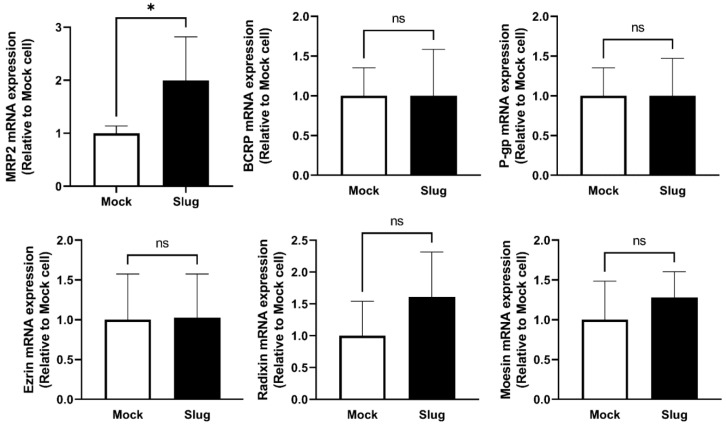
mRNA expression levels of multidrug resistance protein 2 (MRP2), breast cancer resistance protein (BCRP), P-glycoprotein (P-gp), ezrin (Ezr), radixin (Rdx), and moesin (Msn) in Slug-transfected HCC827 cells and Mock-transfected cells. mRNA expression of MRP2 significantly increased after transfection with Slug plasmid (n = 4, ns: not significant, * *p* < 0.05). qRT-PCR data represented the mean ± SD.

**Figure 3 biomolecules-12-00806-f003:**
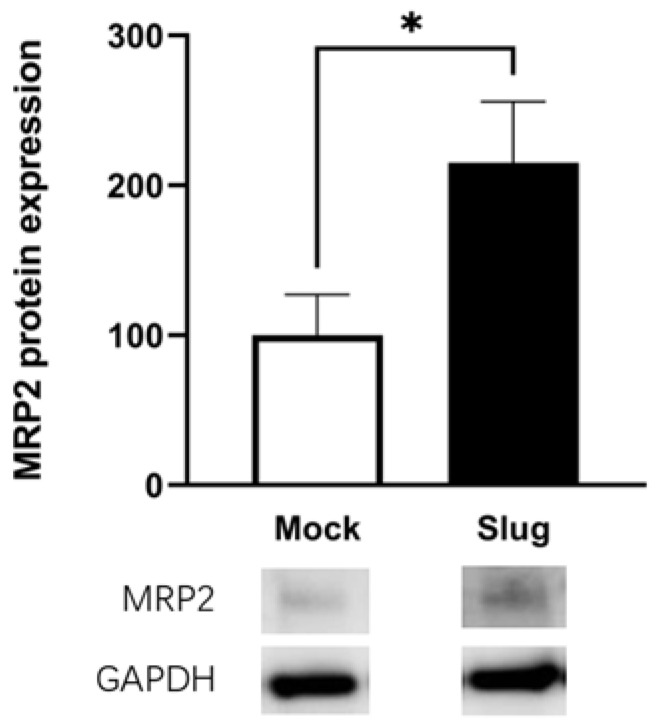
Protein expression levels of MRP2 and Slug in Mock- and Slug-transfected HCC827 cells. The protein expression level of MRP2 was also significantly increased in Slug-transfected cells (n = 4, * *p* < 0.05). Western Blotting data represented the mean ± SD.

**Figure 4 biomolecules-12-00806-f004:**
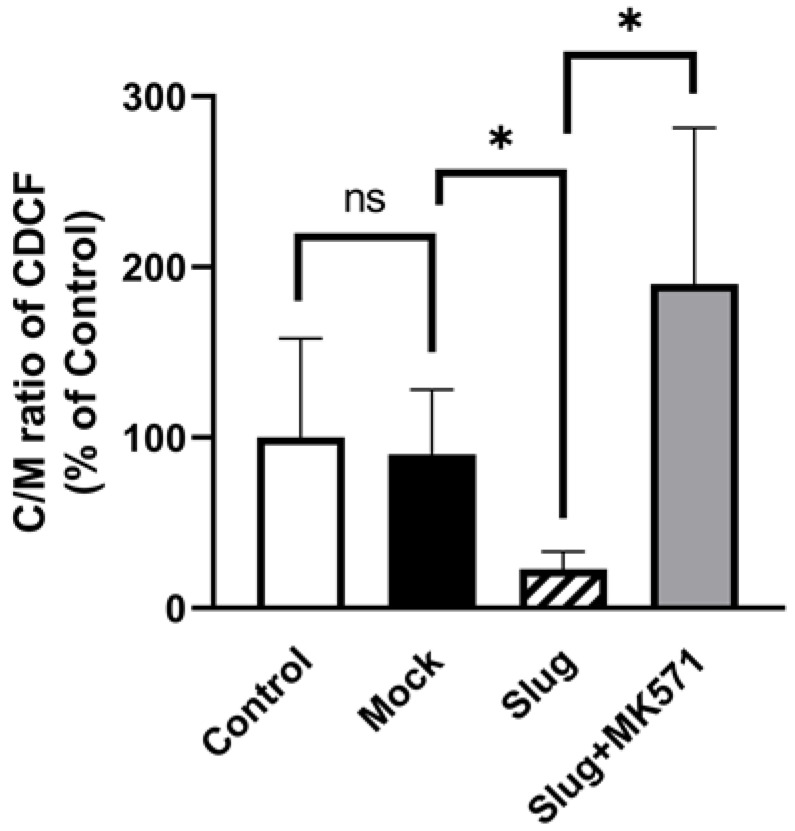
Uptake of CDCF in Slug-transfected HCC827 cells, Mock-transfected cells, and Control cells. All cells were treated with CDCF. Control cells were non-transfected cells. Uptake rate of CDCF in Slug-transfected cells was significantly decreased compared to that in Mock-transfected cells (n = 4, ns: not significant, * *p* < 0.05). However, uptake rate of CDCF was increased when cells were incubated in MK571 for 1 h. Data represented the mean ± SD.

**Figure 5 biomolecules-12-00806-f005:**
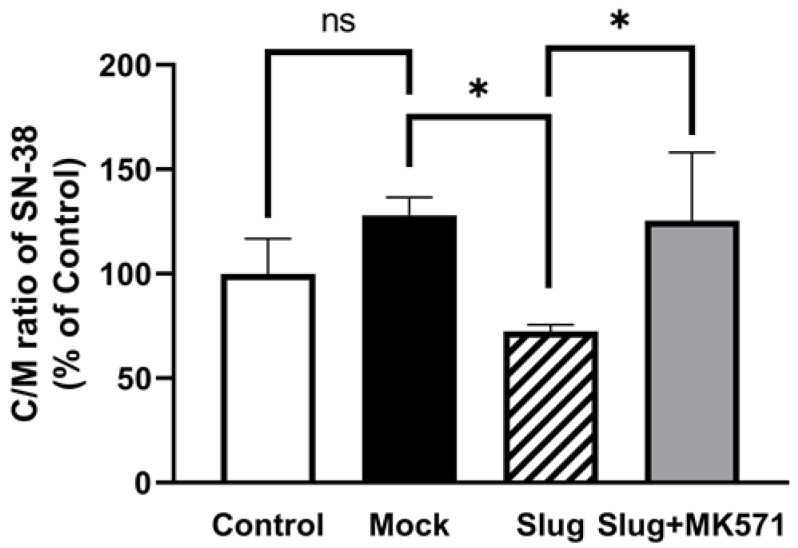
SN-38 uptake assay using 2 μM SN-38. The SN-38 in Slug-transfected cells showed significantly lower uptake than in Mock-transfected cells (n = 3, ns: not significant, * *p* < 0.05). This change was reversed after pre-incubation with 10 μM MK571. Data represented the mean ± SD.

**Figure 6 biomolecules-12-00806-f006:**
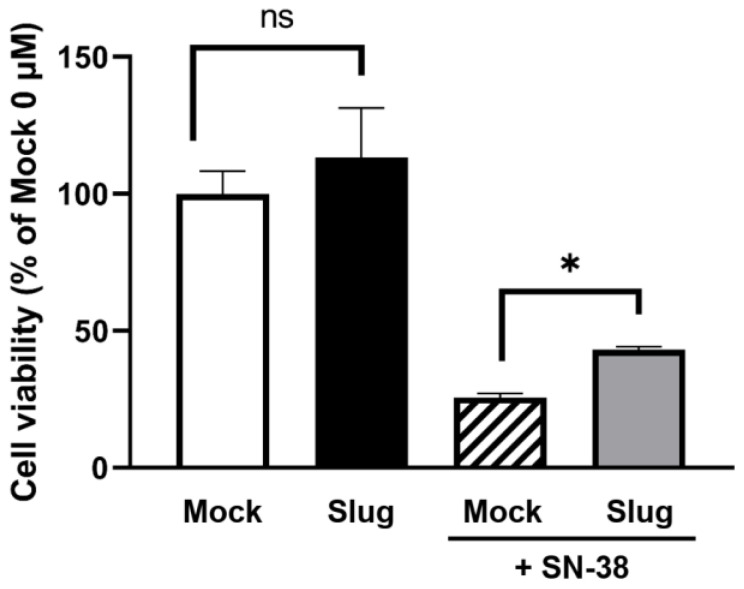
Cell viability assay using 50 μM SN-38. The cell viability of Slug-transfected cells increased significantly (by 38.0%) compared to that of Mock-transfected cells (by 25.7%) (n = 3, ns: not significant, * *p* < 0.05). Data represented the mean ± SD.

## Data Availability

The data presented in this study are available in this article or Appendix A.

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
