# Peer review of "Slug Mediates MRP2 Expression in Non-Small Cell Lung Cancer Cells"

_biomolecules, 2022, doi:10.3390/biom12060806_

Round 1

Reviewer 1 Report

Biomolecules-1729379

The Authors demonstrate the role of Slug in promoting drug resistance in NSCLC cells through the regulation of MRP2 expression. The work is very basic, but rational and well described.

Some major points should be addressed:

  • The title should be modified: the data shown do not demonstrate that scaffold proteins mediate MRP2 expression. Rather, they demonstrate that SLUG affects MRP2 expression.

  • In the abstract: It is not correct to conclude that “Slug may directly increase the mRNA and protein expression….”. As commented below, to claim a direct regulation of Slug, luciferase and ChIP assays should be performed. The data shown, allow to conclude that Slug affect MRP2 expression, but not that it is due to a “direct” control.
  • In Transfection of Slug and Mock plasmids:
  1. How many cells are plated? What kind of wells (24-well plates?)
  2. What is the confluence?
  • In mRNA extraction and cDNA synthesis:
  1. After how many days post-transfection was the RNA extracted?
  2. Gene expression should be calculated by using the formula 2- DDCt (see: Livak KJ, Schmittgen TD. Analysis of relative gene expression data using real-time quantitative PCR and the 2(-Delta Delta C(T)) Method. Methods. 2001 Dec;25(4):402-8 or https://toptipbio.com/delta-delta-ct-pcr/)
  3. A list of the primers used for qRT-PCR must be shown.
  • In “Electrophoresis and western blotting” it is said that protein extraction is performed 6 days post-transfection. Since the cells are transiently transfected, there is no transgene expression after 6 days. For transiently transfected cells, molecular analyses have to be performed not later than 48-72 hrs post-transfection.
  • In “Cell viability assay”:
  1. The “control” has to be the solvent used to dissolve SN-38 and CDCF. Is it DMSO or water?
  2. It is said: “The cells were collected in 96-well plates and the absorbance was measured at 560 nm”. It is not clear. Are the cells trypsinized? They were in 24- well plate and now in 96-well…
  • In “mRNA expression levels of Slug in cells”:
  1. It would be very interesting to see if Slug is expressed in NSCLC samples and if it correlates with resistance in patients.
  2. The expression of Slug at protein level must be shown in Figure 1. Protein extraction should be performed at 48-72 hrs after transfection: at longer times, the transgene would be diluted in the cell population.
  3. In Figure 1, the three panels need to be marked by letters and the legend has to explain what each panel represents.

In general, figure legends do not have to be comments on the results obtained, but description of the figure, how the graph was obtained, what is the control, how many experiments were done, how many replicates, ecc.

In graphs comparing Mock with Slug transfected cells (Figure 1 and Figure 2), it would be better to give to the mock sample the arbitrary value of 1 (this is obtained by dividing the value of 2- DDCt obtained for the Slug sample by the 2- DDCt of the mock sample) and to label the graph with “Fold change” on the y axis. It is also necessary to indicate how many experiments and how many replicates have been performed, or if the graph is a representative experiment performed in duplicate/triplicate.

  • Figure 3:
  1. please provide the original Western Blot of Figure 3 as Supplemental material.
  2. How is the histogram obtained? Please specify in Material and Methods section.
  • Figure 4:
  1. What is the control? If the control is only medium, how comes the C/M ratio is 100?
  2. is the control treated with CDCF?
  3. how is the C/M ratio in Mock cells treated with CDCF and in Mock cells treated with MK571?
  4. Again, the Figure legend lacks this information.
  • Figure 5:
  1. Same comments as in Figure 4.
  • Figure 6:
  1. how many experiments? how many replicates?
  • Line 217:
  1. Authors conclude that Slug directly regulates MRP2 expression. This statement is not supported by experimental data. The direct regulation of a transcription factor on a target gene must be demonstrated by luciferase and ChIP assays.

Minor points:

-In the Abstract:

  1. please write “ERM” in full
  2. Explain what CDCF and SN-38 are

-In Materials:

 in the sentence “Mock and Slug-encoding plasmids were constructed using Vectorbuilder (Chicago, 63 IL, USA) and contained human SNAI2 (NM_003068.5, Slug) or Stuffer_300bp (Mock), re- 64 spectively, ….”, the position of SNAI2 and Stuffer has to be inverted.

- In Electrophoresis and western blotting:

  1. “bands were transferred” is incorrect: rather, proteins were transferred.

- In CDCF uptake assay:

  1. please, when citing CDCF for the first time, indicate the name in full
  2. “Exciation”
  • In 2.9. Cell viability assay
  1. 4 × 104 mL/well should probably be cells/ml
  • Line 167: “remained unchanged”, please add “(not shown)”.

Reviewer 2 Report

In this study, ZHANG et al. focus on the relationships between MRP2 and Slug transcription factor in one lung cancer cell line HCC827.

The manuscript submitted describes a series of results. They use successfully in vitro Slug overexpression studies. The authors observed that Slug-transfected cells showed significantly higher mRNA expression levels of MRP2 than the Mock-transfected cells. Authors observed uptake of both CDCF and SN-38 was significantly decreased after transfection with Slug in HCC827 cells. This change was elegantly abrogated by treatment with MK571, MRP2 inhibitor. However Slug surexpression did not show any significant difference in the mRNA expression levels of ERM protein implied in EMT.

Study would have been more impacting with high soundness if authors confirmed results with another NSCLC cell line.

Reviewer 3 Report

Please find the attachment of comments file

Round 2

Reviewer 1 Report

The manuscript has been impreoved, but important points are still missing.

Title: The term “transcriptional factors” is quite generic…Why don’t “Slug affects MRP2 expression in non-small cell lung cancer”?

Line 15: in the abstract specify that MRP2 is an efflux transporter

Line 26: “Slug may not pass throuth ERM proteins but…” should be better formulated.  “throuth” should be “through”.

Line 135: The authors state: “The parameter Arbitrary to Background (AU-BG) obtained from imaging system was used for statistical analysis”, but they do not specify what was the imaging system used. Please also specify if values are means and if error bars indicate s.d.

Line 156: the authors state that they used 100  μl of DMSO as control in 24- well plates. Since in a single well of a similar multi-well it is possible to pour not more than 500  μl of medium, the volume of DMSO used would be toxic for the cells. The amount of DMSO used should not be higher than 10% and correspond to the amount of DMSO contained in 50 μM SN-38.

Line 158: it is not clear what the sentence “Then the cells were collected by washing out with DMSO to 96-well plates” means. Please explain.

In figure 1, panel D: a graph of protein expression without the correspondent western blot is not acceptable!!

Line 174: “This experiment was performed once”. What experiment? The transfection? Or the qRT-PCR? Please also specify if values are means and if error bars indicate s.d.

Line 187: ”Protein expression level of MRP2 in Slug-overexpression HCC827 cells” should be “ Protein expression level of MRP2 in Slug-overexpressing HCC827 cells”

Figure 4 legend: Please specify that all the cells have been treated with CDCF. Usually the terms “control” and “mock” are referred to cells which have not been treated with the drug.

Line 197: the heading “3.4. Uptake assay” might fit for the heading of the “Materials and Methods” section. Here, in the Results” section, titles such “Slug reduces the uptake of CDCF and SN-38….” Or “Uptake of CDCF and SN-38 in Slug-transfected cells” might be more suitable. The same for line 214 (“3.5. Cell viability assay”)  

Line 203: the sentence “Control cells were defined as the cells without any treatment with plasmids” leads to misunderstanding. It would be better to say: “Control cells are non-transfected cells treated with CDCF”.

Legends of Figure 4, 5 and 6 indicate that these experiments were performed once. Uptake assays, viability assays and in general growth curves need to be performed more than once!

Line 258: “We concluded that Slug directly may not pass through ERM proteins…” should be better formulated.

Reviewer 3 Report

None

Author Response

We thank you so much for helping review our manuscript.